# Cerebral Hemodynamics and Intracranial Compliance Impairment in Critically Ill COVID-19 Patients: A Pilot Study

**DOI:** 10.3390/brainsci11070874

**Published:** 2021-06-30

**Authors:** Sérgio Brasil, Fabio Silvio Taccone, Sâmia Yasin Wayhs, Bruno Martins Tomazini, Filippo Annoni, Sérgio Fonseca, Estevão Bassi, Bruno Lucena, Ricardo De Carvalho Nogueira, Marcelo De-Lima-Oliveira, Edson Bor-Seng-Shu, Wellingson Paiva, Alexis Fournier Turgeon, Manoel Jacobsen Teixeira, Luiz Marcelo Sá Malbouisson

**Affiliations:** 1Division of Neurosurgery, Department of Neurology, Universidade de São Paulo, São Paulo 05403-000, Brazil; swayhs@yahoo.com.br (S.Y.W.); rcnogueira28@gmail.com (R.D.C.N.); marcelodtc@gmail.com (M.D.-L.-O.); edsonshu@hotmail.com (E.B.-S.-S.); wellingson.paiva@hc.fm.usp.br (W.P.); manoeljacobsen@gmail.com (M.J.T.); 2Department of Intensive Care, Universitè Libre de Bruxelles, 1000 Brussels, Belgium; ftaccone@ulb.ac.be (F.S.T.); filippo.annoni@erasme.ulb.ac.be (F.A.); 3Department of Intensive Care, Universidade de São Paulo, São Paulo 05403-000, Brazil; bmtomazini@gmail.com (B.M.T.); sergio.fonseca@hc.fm.usp.br (S.F.); estbassi@gmail.com (E.B.); bruno.lucena@hc.fm.usp.br (B.L.); luiz.malbouisson@hc.fm.usp.br (L.M.S.M.); 4Division of Critical Care Medicine and the Department of Anesthesiology, Université Laval, Québec City, QC G1V 0A6, Canada; alexis.turgeon@fmed.ulaval.ca

**Keywords:** intracranial pressure, intracranial compliance, cerebrovascular resistance, cerebral perfusion pressure, COVID-19

## Abstract

*Introduction*: One of the possible mechanisms by which the new coronavirus (SARS-Cov2) could induce brain damage is the impairment of cerebrovascular hemodynamics (CVH) and intracranial compliance (ICC) due to the elevation of intracranial pressure (ICP). The main objective of this study was to assess the presence of CVH and ICC alterations in patients with COVID-19 and evaluate their association with short-term clinical outcomes. *Methods*: Fifty consecutive critically ill COVID-19 patients were studied with transcranial Doppler (TCD) and non-invasive monitoring of ICC. Subjects were included upon ICU admission; CVH was evaluated using mean flow velocities in the middle cerebral arteries (mCBFV), pulsatility index (PI), and estimated cerebral perfusion pressure (eCPP), while ICC was assessed by using the P2/P1 ratio of the non-invasive ICP curve. A CVH/ICC score was computed using all these variables. The primary composite outcome was unsuccessful in weaning from respiratory support or death on day 7 (defined as UO). *Results*: At the first assessment (*n* = 50), only the P2/P1 ratio (median 1.20 [IQRs 1.00–1.28] vs. 1.00 [0.88–1.16]; *p* = 0.03) and eICP (14 [11–25] vs. 11 [7–15] mmHg; *p* = 0.01) were significantly higher among patients with an unfavorable outcome (UO) than others. Patients with UO had a significantly higher CVH/ICC score (9 [8–12] vs. 6 [5–7]; *p* < 0.001) than those with a favorable outcome; the area under the receiver operating curve (AUROC) for CVH/ICC score to predict UO was 0.86 (95% CIs 0.75–0.97); a score > 8.5 had 63 (46–77)% sensitivity and 87 (62–97)% specificity to predict UO. For those patients undergoing a second assessment (*n* = 29), after a median of 11 (5–31) days, all measured variables were similar between the two time-points. No differences in the measured variables between ICU non-survivors (*n* = 30) and survivors were observed. *Conclusions*: ICC impairment and CVH disturbances are often present in COVID-19 severe illness and could accurately predict an early poor outcome.

## 1. Introduction

The severity of the disease caused by the new coronavirus 2019 (COVID-19) is predominantly harbored in the occurrence of severe acute hypoxemic respiratory failure, which often requires ventilatory support. However, the involvement of other organs, such as the central nervous system (CNS), heart, kidneys, intestines, and testicles [1,2,3,4,5,6], has been reported. In particular, neurological manifestations vary from acute cerebrovascular events to immuno-mediated diseases, such as Guillain–Barre syndrome or cytotoxic lesion of the corpus callosum [7,8,9]. Importantly, it remains still unclear whether CNS disorders are the consequences of acute respiratory failure, systemic aggressions, or the hypercoagulable status, or are secondary to the primary CNS invasion from the virus [10].

As many COVID-19 patients experience neurological symptoms, such as headache, anosmia, paresthesia, nausea, vomiting, and alteration of consciousness [11], one hypothesis could also be that intracranial compliance (ICC) and cerebrovascular hemodynamics (CVH) are impaired in the early course of the disease, either directly (i.e., encephalitis, brain edema, or focal ischemia) or indirectly (i.e., hypoxic distress, cytokine storm, and endothelial dysfunction) [12,13]. This would be even more frequent in critically ill patients, who often suffer from persistent somnolence, lethargy, and delirium [14,15].

Intracranial pressure (ICP) monitoring is the cornerstone of the clinical management of acute brain-injured patients at risk of intracranial hypertension (ICH) [16]. Still, this monitoring tool has limited use outside of neuro-critical settings because of the invasive nature of the monitoring and its availability. ICH would result in altered ICC and, consequently, secondary brain damage. In COVID-19 patients, unless structural damage with mass effect is documented on brain imaging, there is no indication to implant an ICP monitor; however, non-invasive techniques able to estimate CVH and/or ICP curves [17,18,19] may be helpful to understand changes in cerebral perfusion in this setting [20].

The purpose of the study was therefore to evaluate the relationship between CVH and ICC and clinical outcomes in a cohort of mechanically ventilated critically ill COVID-19 patients.

## 2. Methods

### 2.1. Study Design

We conducted a single center, observational, prospective study in six intensive care units (ICUs) of Hospital das Clínicas, São Paulo University, Brazil, from May to June 2020; during the first wave of the pandemic, the entire hospital, including ICUs, was dedicated to severe COVID-19 management. This clinical trial (CT) study protocol was approved by the Research Ethical Committee at Sao Paulo University Medical School, on 19 April 2020 (REB register 31750820.1.0000.0068) and registered under number NCT04429477 (available at clinicaltrials.gov, accessed on 9 June 2020). All methods were performed in accordance with the relevant guidelines and regulations, and informed consent was obtained from all legally authorized representatives or next of kin.

All consecutive patients with confirmed COVID-19 by real-time reverse transcription–polymerase chain reaction positive testing were eligible within the first 72 h of the initiation of invasive mechanical ventilation. Exclusion criteria included the absence of a temporal acoustic window for TCD assessment, the absence of a dedicated operator for ICC and CVH assessment, patients unable to undergo ICC monitoring due to lesions and/or skin infections in the sensor application region, and patients with head circumference smaller than 47 cm. The study protocol was according to the Standards for Reporting of Diagnostic Accuracy Studies (STARD) statement.

Eligible subjects were identified by the ICU teams (SYW, SF, BT, EB, and LMSM). Two assessments of CVH and ICC were performed: the first during the first three days from intubation and the second up to 72 h after extubation or tracheotomy without administration of sedatives; for patients who died while intubated, only the first evaluation was considered; ICC and TCD recordings lasted 30 min for each session and were performed by the same operator, who was unaware of the clinical features and characteristics of the patient. Clinical parameters, such as systemic arterial pressure, fluid balance, use of sedatives, PaO_2_ and PaCO_2_, hemoglobin, and body temperature, were concomitantly recorded. Data on demographic characteristics, simplified acute physiologic score (SAPS) 3, use of intravenous sedatives, vasopressors, and other physiological and laboratory data were also collected. As the ICC and CVH assessment was exploratory, all results were not available to the treating ICU physicians and therefore did not influence clinical decisions and therapeutics.

### 2.2. Intracranial Compliance Monitoring Technique

ICC was evaluated non-invasively by assessing cranial deformation using a specific device (B4C; Brain4care Corp., São Carlos, Brazil). The B4C sensor consists of a support for a sensor bar that detects local cranial bone deformations using specific sensors. The detection of these deformations is obtained by a cantilever bar modeled through finite element calculations. Voltage meters are attached to this bar for deformation detection. Non-invasive contact with the skull is obtained by adequate pressure directly into the scalp by means of a pin. The system is positioned in the frontotemporal region, around 3 cm over the first third of the orbitomeatal line; consequently, avoiding temporal superficial artery main branches and temporal muscle, providing contact of the sensor with an area of thin skin and skull, whereas slight pressure is applied to the adjustable band until the optimal signal is detected.

Variations in ICP cause deformations in the cranial bone, which are detected by the sensor bar. The device filters, amplifies, and scans the sensor signal and sends the data to a mobile device. The method is completely non-invasive and painless. In addition, it does not interfere with any routine monitoring. The waveform obtained is equivalent to the ICP waveform obtained using invasive techniques, such as intraparenchymal probes or external ventricular drain [21], and the relation between its different components provides information on ICC [22]. In particular, each cardiac beat corresponds to an ICP waveform composed of three peaks: arterial pulsation (P1); cerebral venous flow, which is secondary to cyclic fluctuations of arterial blood volume, reflecting intracranial compliance (P2); and the aortic valve closure (P3; Figure 1) [23].

The B4C analytics system verified all data collected by the sensor, i.e., ICP pulse waves morphology parameters such as the P2/P1 ratio. For this study, all calculations were performed using the mean pulse of the ICP, calculated by identifying and extracting all ICP pulses, excluding possible artifacts. The mean pulse was used to calculate the amplitudes of the P1 and P2 peaks, which were obtained by detecting the highest point of these peaks and subtracting the base value of the ICP pulse. The P2/P1 ratio was calculated by dividing the amplitude of these two points. In the case of P2 > P1, ICC was defined as “abnormal”.

### 2.3. Cerebrovascular Hemodynamics Assessment

Conventional transcranial Doppler (TCD; EZ-DOP, DWL Compumetrics, Singen, Germany) was used to assess CVH [24]. A complete evaluation of right and left cerebral hemispheres and the brainstem arteries was performed prior to the study to discard focal stenosis, using the Doppler colored technique with a low-frequency probe (2MHz) and scanning every 1 mm of arterial extension through the temporal, orbital, suboccipital, retro-mastoid, and submandibular windows. Hemodynamic parameters of interest were mean flow velocities in the middle cerebral arteries (mCBFV) and peak systolic and diastolic velocities, as MCAs supply blood flow to the largest areas of the brain hemispheres [25]. Abnormal mCBFV was identified by values <40 or ≥100 cm/s [26].

Using TCD, elevation of ICP was suspected when pulsatility index (PI) ≥1.2 (i.e., “abnormal” PI) [27]. PI was calculated by the following formula: PI = Sv − Dv/Mv (Sv: systolic velocity, Dv: diastolic velocity, and Mv: mean flow velocity). Moreover, TCD allows the calculation of estimated CPP (eCPP) and ICP (eICP) [28], which are significantly correlated with invasive ICP measurements [24,29]. Abnormal eICP was considered if >20 mmHg, abnormal eCPP if ≤45 or ≥75 mmHg.

### 2.4. Outcomes

As a wide range of variables is involved in the prognosis of COVID-19 [30,31,32,33], we limited our analyses to the prevalence and predictive values of ICC and CVH disturbances on an early unfavorable outcome (UO); UO was a composite end-point including either the absence of weaning from mechanical ventilation (MV) or death on day 7 after inclusion in the study.

CVH and ICC impairment were identified using the different combinations of TCD and B4C values; in particular, P2/P1 ratio, mCBFV, eICP, PI, and eCPP were categorized and an arbitrary score was developed to describe different degrees of these alterations (Table 1), already applied in a previous published study [34]. For each variable, severity was defined by a CVH/ICC score from 1 to 4. As such, the sum of the severity score for each variable gave a score ranging from a minimum of 5 to a maximum of 20. The score was then classified as “normal”, i.e., 5 points, which suggested no abnormalities; “mild CVH/ICC abnormalities”, i.e., 6 to 7 points, which was associated with minor disturbances in one or two variables; “moderate CVH/ICC abnormalities”, i.e., 8 to 9 points; and “severe CVH/ICC impairment”, i.e., ≥10 points.

### 2.5. Sample Size

Due to the lack of clinical reports in this field, this research should be considered a pilot study. Using the upper confidence interval for the population variance approach to the sample size calculation, a pilot sample size between 20 and 40 was chosen, corresponding to standardized effect sizes of 0.4 and 0.7 (for 90% power based on a standard sample size calculation) [35]. Thus, considering the risk of early deaths and lack of second TCD and B4C assessment, 50 patients were enrolled to test our hypothesis.

### 2.6. Statistical Analysis

Descriptive statistics were computed for all study variables. Categorical variables are presented as count (%), while continuous variables are presented as mean (±standard deviation) or median (25th–75th percentiles), according to their distribution. Differences between groups (i.e., UO vs. favorable outcome; non-survivors vs. survivors) were assessed using a χ-square or Fisher’s exact test for categorical variables, t-Student test for normally distributed continuous variables, and Mann–Whitney or Kruskal–Wallis tests for asymmetrically distributed continuous variables. Differences between the first and second assessments were assessed using a Wilcoxon paired test. The discriminative ability of each variable (i.e., P2/P1, mCBFV, PI, eICP, and eCPP) with a significant difference between UO and favorable outcome was evaluated to predict UO, using receiver operating characteristic (ROC) curves with the corresponding area under the curve (AUROC). Youden’s index was computed to assess the optimal predictive cut-off. A *p* < 0.05 was considered statistically significant. All analyses were performed using the AcaStat software version 2200.5.2 (AcaStat, Winter Garden, FL, USA, available at www.acastat.com, accessed on 15 January 2021).

## 3. Results

### 3.1. Study Population

Between May and June 2020, 2813 patients with COVID-19 were admitted to our institution. Among those, 1579 were admitted to ICU and 552 critically ill patients died (35%). As the availability of personnel to perform TCD and B4C assessments was scarce, and considering the exploratory purpose of the study, a total of 50 patients were eventually included within 72 h of ICU admission. Characteristics of the study population are shown in Table 2. On day 7 from inclusion into the study, 31 (62%) were still on MV and 4 (8%) had died, resulting in 35 (70%) with UO. After a median of 11 (5–31) days, 29 patients (18 of which still required intermittent non-invasive ventilation) were weaned from MV and underwent a second CVH/ICC assessment. At ICU discharge, 2 of the 15 patients with FO on day 7 had died; 24 out of the 31 alive on day 7 but still on MV (UO) eventually died during the ICU stay. As such, ICU mortality was observed in 30 (60%) patients (Table 2).

### 3.2. ICC and CVH Assessment

At the first assessment (*n* = 50), mCBFV was 63 (32–140) cm/s, PI was 1.04 (0.65–2.2), eICP 16 (1–48) mmHg, and eCPP 63 (30–90) mmHg. Twenty-six (52%) patients had abnormal mCBFV; low mCBFV (<40 cm/s) was observed in five patients and elevated mCBFV was observed in 21 patients. Elevated PI was observed in 19 (38%) patients, high eICP in 14 (28%) patients, and abnormal eCPP in 11 (22%) patients (four with low eCPP and seven with high eCPP). Thirty-three (66%) patients presented abnormal P2/P1 ratio (16 within 1.01–1.2 and 17 > 1.2—Figure 2). The CVH/ICC score was 8.5 (7–11.75).

At the second assessment (*n* = 29), mCBFV was 68 (30–173) cm/s, PI was 0.99 (0.5–1.8), eICP 15 (3–31) mmHg, and eCPP 71 (41–97) mmHg. Twelve (41%) patients had abnormal mCBFV; low mCBFV (<40 cm/s) was observed in two patients, and elevated mCBFV (>70 cm/s) was observed in 10 patients. Elevated PI was observed in 10 (34%) patients, high eICP in seven (24%) patients, abnormal eCPP in 14 (48%) patients (2 low and 12 high), and abnormal P2/P1 ratio in 15 (51%) patients (4 for 1.01–1.2 and 11 for >1.2).

For these 29 patients, P2/P1 (1.05 [0.96–1.27] vs. 1.08 [0.87–1.28]; *p* = 0.86), mCBFV (73 [53–82] vs. 60 [51–80] cm/s; *p* = 0.27), PI (0.95 [0.85–1.27] vs. 0.92 [0.86–1.21]; *p* = 0.58), eICP (13 [9–21] vs. 14 [10–20] mmHg; *p* = 0.96), and eCPP (68 [60–75] vs. 75 [60–85] mmHg; *p* = 0.09) were similar between the two assessments. Table 3 shows the concordance between abnormal values for the different parameters measured at the first assessment. Most patients who presented abnormal eICP also had abnormal PI or P2/P1 values. For those with both available assessments, no significant differences were observed between the two measurements within patients with favorable and unfavorable outcome. The CVH/ICC score was 8 (6–10) at the second assessment. The CVH/ICC score was similar at the two time-points (*p* = 0.48) for the 29 patients with two assessments.

### 3.3. CVH/ICCI and Primary Outcome

At the first assessment (*n* = 50), mCBFV (66 [43–80] vs. 63 [53–73] cm/s; *p* = 0.70), PI (1.01 [0.90–1.40] vs. 0.95 [0.85–1.10]; *p* = 0.13), and eCPP (64 [56–71] vs. 64 [60–74] mmHg; *p* = 0.35) values were similar between patients with UO and favorable outcome. However, P2/P1 (1.20 [1.00–1.28] vs. 1.00 [0.88–1.16]; *p* = 0.03) and eICP (14 [11–25] vs. 11 [7–15] mmHg; *p* = 0.01) were significantly higher among patients with UO than the others.

In addition, the proportion of patients with abnormal mCBFV (21/35, 60% vs. 5/15, 33%, OR 3.00 [0.90–9.97]; *p* = 0.08), PI (16/35, 46% vs. 3/15, 20%, OR 3.36 [0.89–12.45]; *p* = 0.09), eICP (12/35, 34% vs. 2/15, 13%, OR 3.39 [0.71–16.75]; *p* = 0.13), eCPP (9/35, 26% vs. 2/15, 13%, OR 2.29 [0.42–11.43]; *p* = 0.33), and P2/P1 values (26/35, 74% vs. 7/15, 47%, OR 3.30 [0.95–11.36]; *p* = 0.06) were similar between groups.

Patients with UO had a significantly higher CVH/ICC score (9 [8–12] vs. 6 [5–7]; *p* < 0.001) than those with a favorable outcome (Figure 3). The AUROC for CVH/ICC score to predict UO was 0.86 (95% CIs 0.75–0.97); a score >8.5 had 63 (46–77)% sensitivity and 87 (62–97)% specificity to predict UO. One out of five (20%) patients with no CVH/ICC impairment had UO; 7/15 (46%) patients with mild CVH/ICC impairment had UO; 10/12 (83%) patients with moderate CVH/ICC impairment had UO; 17/18 (94%) patients with severe CVH/ICC impairment had UO (*p* = 0.001).

### 3.4. CVH/ICCI and ICU Mortality

At the first assessment (*n* = 50), mCBFV (60 [42–80] vs. 72 [54–80] cm/s; *p* = 0.16), PI (1.00 [0.92–1.37] vs. 0.96 [0.86–1.29]; *p* = 0.26), eICP (14 [9–23] vs. 13 [8–17] mmHg; *p* = 0.48), eCPP (64 [54–70] vs. 66 [60–75] mmHg; *p* = 0.12), and P2/P1 values (1.14 [0.93–1.25] vs. 1.08 [0.96–1.27]; *p* = 0.84) were similar between non-survivors and survivors. Additionally, the proportion of patients with abnormal mCBFV, PI, eICP, eCPP, and P2/P1 values was similar between groups, as the CVH/ICC score (9 [7–12] vs. 7 [6–11]; *p* = 0.13).

## 4. Discussion

In this single-center prospective study, we observed that brain hemodynamics were impaired in a large proportion of critically ill COVID-19 patients on mechanical ventilation. Early assessment of these patients showed higher intracranial pressure and lower intracranial compliance among those who were still on ventilation or who had died within the first 7 days after study admission. Moreover, a score composed of different parameters related to brain perfusion and compliance could identify such patients with high accuracy. Nevertheless, these alterations were not associated with ICU mortality.

ICC is a concept that is rarely considered during critical illnesses, mostly because of the lack of techniques available to assess it [34]. ICP invasive monitoring in COVID-19 patients would have been unethical and difficult to implement. The B4C device does not provide ICP values; however, it allows a bedside observation and report of the ICP pulse waveform quantitative parameters, a vital sign that is gaining attention in recent years [36,37]. Studies evaluating this ICC device in critically ill patients are few and still ongoing [38]; nevertheless, ICC was adequately assessed in children with hydrocephalus [21] and in experimental studies using this technique [39]. Together with TCD-derived variables, a combination of cerebral hemodynamics and intracranial compliance assessment could provide interesting information about brain function in different critically ill patients [20]; in particular, in our study, we observed that only the CVH/ICC score was significantly different between FO and UO patients, which may suggest that the combination of these parameters, rather than a single one, could be more sensitive and a better predictor for patients’ outcome in this setting. Importantly, further studies are required in this field, as different methods for non-invasive ICP assessment have different precision, and concordance among them needs to be adequately assessed. Additionally, none of these tools could precisely replace the use of invasive ICP monitoring, which remains the cornerstone of managing severely brain-injured patients currently.

Alterations in intracranial hemodynamics in COVID-19 patients have been previously reported only in two studies [40,41], which did not specifically assess brain compliance, whereas several studies have shown an increased occurrence of cerebrovascular events in low-risk patients, being associated with coagulation disorders and/or the formation of clots into the intracranial vessels or the brain microvasculature [42]. Nervous system injury associated with COVID-19 has been associated with a direct viral penetration of nerve endings (i.e., traveling within the axons) or by the systemic circulation (i.e., via the infected endothelial cells of brain vessels or epithelial cells in the choroid plexus) or via the disruption of the blood-brain barrier. Additionally, neuroinflammation, which is triggered by the systemic inflammatory response and organ damage, can contribute to brain dysfunction through the activation of microglial cells [43]. The clinical consequences of these phenomena are consistent with cerebrovascular events, brain swelling, seizures, and encephalopathy, which could be further enhanced by the use of specific therapies, such as mechanical ventilation, sedatives, or extra-corporeal membrane oxygenation (ECMO) [44].

The role of TCD-derived parameters (mCBFV, PI, eICP, and eCPP) for the assessment of cerebral hemodynamics in critically ill patients without a primary brain injury is well established, with relevant data on the role of altered autoregulation in the pathogenesis of septic encephalopathy [45] or of reduced mCBFV in association with severe post-anoxic brain damage [46], although the association with long-term neurological outcome needs to be further evaluated. In hospitalized COVID-19 patients, higher mCBV and lower vasoreactivity were observed than in matched healthy volunteers [47]. Additionally, in a small recent study using cerebral ultrasound, eICP was higher and diastolic CBFV lower in COVID-19 patients developing neurological complications when compared to others [40]. The authors also added the estimation of ICP using the optic nerve sheath diameter (ONSD) measurement with cerebral ultrasound, which was not available in our study. Moreover, mortality was higher in our study than in that report (60% vs. 33%), and differences in the selection and management of patients as well as a different study endpoint precluded any additional comparisons.

We did not specifically evaluate which factors could influence alterations in brain hemodynamics or compliance. Changes in PaCO_2_, mean arterial pressure, pH, sodium, or temperature could also lead to altered brain hemodynamics [48,49,50]. Because of the relatively small sample size and different patterns of TCD and P2/P1 alterations, we could not assess the association of systemic abnormalities with some specific patterns, such as low mCBFV, high eICP, or P2/P1. However, as all these variables are complementary and evaluate different aspects of brain hemodynamics, we could obtain a composite derived score with a high predictive accuracy for early UO. Further studies would be required to confirm the validity of this score and its association with clinically relevant neurologic symptoms or syndromes (i.e., encephalopathy, delirium, ischemic events) or its correlation with other neuromonitoring tools, such as brain oximetry or electroencephalography.

This study has several limitations to be acknowledged. First, our findings showed association and not causality between UO and alterations in cerebral hemodynamics. Routine daily assessment of these variables as potential therapies to restore “normal” brain hemodynamics should be further evaluated in this setting. Second, data on the ventilatory parameters such as PEEP, tidal volume, and plateau pressure were not available for all patients. However, early in the COVID-19 pandemic in Brazil, when the study was performed, the institutional protocol for mechanical ventilation in COVID-19 was based on the ARDSNet protocol [51], which recommended using a tidal volume of 6 mL/kg of ideal body weight and setting the PEEP level according to the FiO_2_. Given the similarities between groups in the P/F ratio, it is reasonable to consider that other ventilatory parameters were quite similar between groups. Third, we could not perform neurological imaging during the study period, which would have been necessary to elucidate the etiology of the alterations in intracranial pressure pulse waveform in COVID-19, whether as primary CNS injury or secondary to respiratory or other systemic complications. Fourth, the CVH/ICC score was arbitrarily created and needs to be further validated in larger cohorts of critically ill patients and, if possible, correlated to other predictors (i.e., imaging, biomarkers, electrophysiological testing) of brain injury. Finally, the availability of the operator to assess brain hemodynamics was limited for the study, which resulted in a small cohort of patients included from a large eligible population of severe COVID-19 patients, potentially resulting in a selection bias.

## 5. Conclusions

In this study, early alteration of brain hemodynamics and compliance were associated with the severity of COVID-19, including death or dependency from mechanical ventilation. These findings advocate for larger investigations on the role of neuromonitoring seen in this setting to further understand the role of such disturbances on the outcome of clinically relevant patients.

## Figures and Tables

**Figure 1 brainsci-11-00874-f001:**
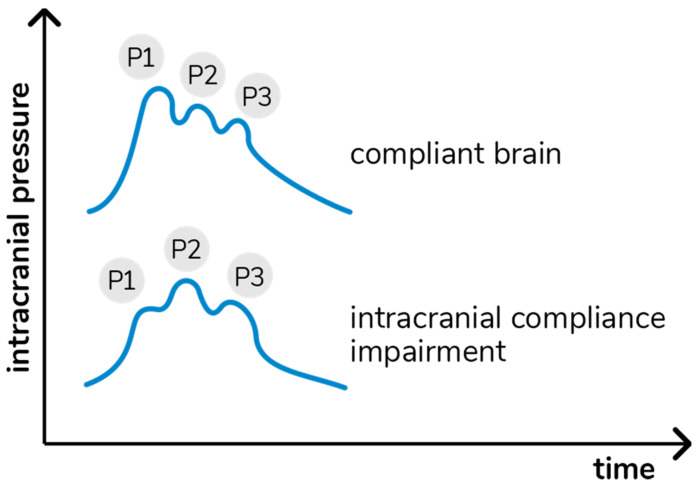
Intracranial pressure waves morphology in accordance with cerebral compliance.

**Figure 2 brainsci-11-00874-f002:**
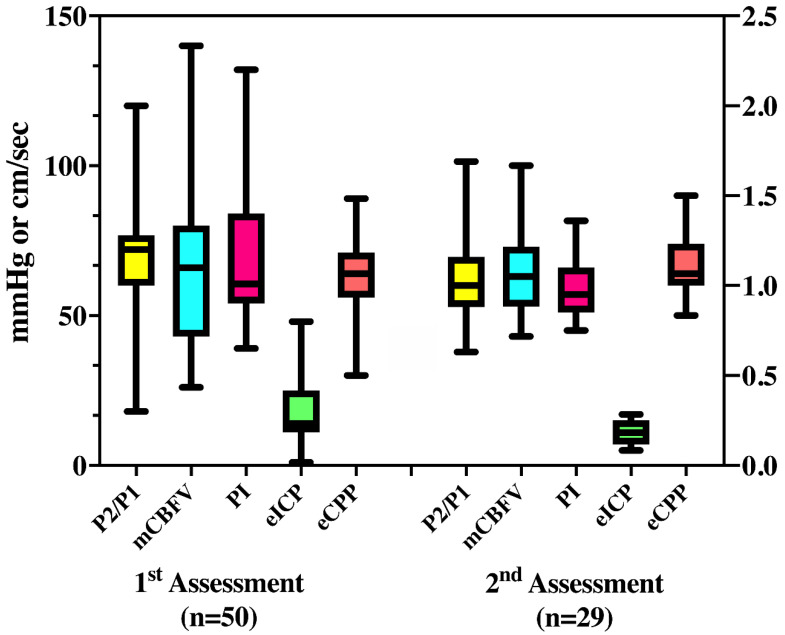
Median values of all variables derived from transcranial doppler and cerebral compliance (P2/P1) measurement, on the first and second assessment. mCBFV = mean cerebral blood flow velocity (cm/sec); PI = pulsatility index; eICP = estimated intracranial pressure (mmHg); eCPP = estimated cerebral perfusion pressure (mmHg).

**Figure 3 brainsci-11-00874-f003:**
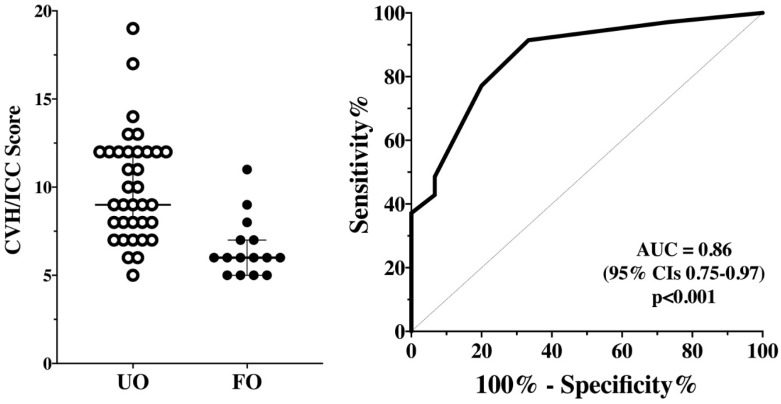
CVH/ICC score between patients with unfavorable (UO) and favorable outcomes (FO) on the first assessment (*n* = 50—**Left Panel**). The area under the receiver operating characteristic (ROC) for CVH/ICC score to predict UO is shown on the **Right Panel**.

**Table 1 brainsci-11-00874-t001:** Thresholds for P2/P1 ratio, mCBFV, PI, eICP and eCPP. Progressive points were in accordance with the worst results. CVH/ICCI: cerebrovascular hemodynamics and intracranial compliance impairment, mCBFV: middle cerebral artery highest mean velocity; eICP: estimated intracranial pressure; eCPP: estimated cerebral perfusion pressure.

Points	P2/P1	mCBFV	PI	eICP	eCPP	Score (Sum of Each)
1	≤1	40–70	<1.2	<15	50–75	5 no CVH/ICCI
2	1.01–1.19	71–99	≥1.2	15–20	≥75	6–7 mild CVH/ICCI
3	≥1.2	≥100	≥1.3	21–25	≤50	8–9 moderate CVH/ICCI
4	≥1.4	<40	≥1.4	>25	<40	≥10 severe CVH/ICCI

**Table 2 brainsci-11-00874-t002:** Clinical characteristics of patients with COVID-19, according to the occurrence of the primary outcome (FO: favorable outcome, UO: unfavorable outcome). * *p* < 0.05 FO vs. UO or survivors vs. non-survivors.

Characteristic	Total (*n* = 50)	FO (*n* = 17)	UO (*n* = 33)	Survivors (*n* = 20)	Non-Survivors (*n* = 30)
Age, median (IQR)—years	62 (44–68)	38 (17–63)	64 (49–68)	49 (17–63)	66 (51–68) *
Female gender—no. (%)	22 (44)	8 (48)	14 (43)	9 (43)	13 (45)
SAPS 3, median (IQR)	63 (52–75)	55 (51–72)	69 (57–78)	58 (51–69)	67 (54–78)
Antibiotics use—no. (%)	36 (77)	14 (70)	22 (82)	15 (71)	21 (81)
Comorbidities and risk factors—no. (%)					
Hypertension	26 (54)	9 (52)	17 (51)	11 (52)	15 (56)
Diabetes	15 (31)	6 (35)	9 (27)	6 (29)	9 (33)
Obesity	23 (46)	9 (52)	14 (42)	9 (43)	11 (42)
Current smoker	15 (31)	5 (29)	10 (30)	6 (29)	9 (33)
Chronic kidney failure	7 (15)	2 (11)	5 (15)	2 (10)	5 (19)
Laboratory variables					
D-dimer—ng/mL, median (IQR)	2304 (1120–5528)	2018 (1120–6027)	2512 (1007–18,631)	2038 (1166–4792)	2449 (1007–18,631)
Creatinine—mg/dL, median (IQR)	1.25 (0.66–2.71)	1.23 (0.66–2.52)	1.41 (0.66–4.59)	0.98 (0.62–1.82)	1.45 (0.69–4.59)
Platelets count—×10^9^/L	240 (150–333)	293 (146–375)	217 (146–310)	257(170–375)	226 (146–310)
Intravenous sedation					
Midazolam—no. (%)	44 (88)	12 (70)	23 (69)	18 (86)	26 (90)
Propofol—no. (%)	13 (26)	6 (30)	7 (21)	7 (33)	6 (21)
Ketamine—no. (%)	2 (4)	0	2 (6)	0 (0)	2 (7)
On the day of the first assessment					
MAP—mmHg	79 (72–87)	83 (75–90)	77 (72–83)	84 (75–90)	78 (72–82)
MAP < 65 mmHg—no. (%)	2 (4)	–	2 (6)	–	2 (7)
Vasopressor use—no. (%)	25 (50)	9 (50)	16 (32)	8 (38)	17 (77)
Heart rate—mmHg	87 (75–98)	75 (72–81)	87 (79–104)	78 (72–93)	89 (82–104)
Respiratory rate—rpm	26 (22–30)	25 (22–31)	29 (23–32)	26 (22–29)	28 (22–32)
Oxygen saturation (SaO_2_)—%	93 (90–95)	94 (90–97)	92 (90–95)	93 (91–94)	92 (90–95)
PaO_2_ < 60 mmHg—no. (%)	3 (6)	1 (5)	2 (6)	1 (5)	2 (7)
PaCO_2_—mmHg	41 (37–49)	39 (36–44)	42 (40–44)	40 (37–44)	42 (38–44)
PaCO_2_ > 45 mmHg)—no. (%)	16 (32)	4 (22)	12 (37)	5 (24)	11 (38)
Temperature—°C	36.1 (35.6–36.9)	36 (35.8–37)	36.5 (35.5–38)	36.2 (35.5–37.0)	36.2 (35.7–38.0)
Hemoglobin—g/dL	11.1 (8.8–12.7)	11.2 (9.9–12.9)	10.3 (8.7–13.1)	11.6 (10.4–12.4)	10.2 (8.7–13.1)
PaO_2_/FiO_2_ –median (IQR)	145 (127–182)	178 (141–216)	132 (98–151)	167 (138–216)	136 (98–151) *
Renal replacement therapy—no. (%)	13 (32)	5 (27)	8 (25)	5 (26.3)	8 (36)

MAP = mean arterial pressure.

**Table 3 brainsci-11-00874-t003:** Concordance of abnormal values for each technique on the first assessment (*n =* 50).

	Abnormal mCBFv *n =* 26	Abnormal PI *n =* 19	Abnormal eICP *n =* 14	Abnormal eCPP *n =* 11	Abnormal P2/P1 *n =* 33
Abnormal mCBFV *n =* 26	-	11 (42)	10 (38)	7 (27)	18 (69)
Abnormal PI *n =* 19	11 (58)	-	13 (68)	5 (26)	14 (73)
Abnormal eICP *n =* 14	10 (71)	13 (93)	-	6 (42)	13 (93)
Abnormal eCPP *n =* 11	7 (63)	5 (45)	6 (54)	-	6 (54)
Abnormal P2/P1 *n =* 33	18 (54)	14 (42)	13 (39)	6 (18)	-

## Data Availability

Datasets used and/or analyzed during the current study are available from the corresponding author on reasonable request.

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
