# Peer review of "Cerebral Hemodynamics and Intracranial Compliance Impairment in Critically Ill COVID-19 Patients: A Pilot Study"

_brainsci, 2021, doi:10.3390/brainsci11070874_

Round 1
Reviewer 1 Report
I read Brasil et al’s article with great interest. The authors used TCD and B4C to assess cerebral vascular hemodynamics (CVH) and intracranial compliance (ICC) for patients with COVID-19 and they intend to evaluate the relationship between CVH and ICC in a cohort of mechanically ventilated critically ill patients infected with SARS-CoV-2 on clinical outcomes. They concluded that early deteriation of cerebral hemodynamics and compliance were associated with the severity of COVID-19 patients, including dependency on mechanical ventilation and death. The article is well written and organized. The data are newly collected from recent COVID-19 patients and the results or conclusion is extremely helpful for COVID-19 patient treatment in ICU. I only have several minor comments as below:
- Convid-19 was used through the whole article, except the last sentence of the introduction section, where they used SARS-CoV-2. Could they make the terminology consistent please?
- Abstract: Please describe the data format in the abstract. For example, 1.20 [1.00-1.28], I assume the data format here is median [95% CI].
- Abstract: The last sentence: ICCI? It should be ICC?
- Delete B4C device in the abstract. It is unnecessary.
- Methods: this study includes data from six intensive ICUs, I assume these are different ICUs? Some are cardiac ICU? Some are Neuro ICU or PCIU? Can they give more details please?
- Methods: Can they clarify whether the measurement of TCD or B4C influence clinical decisions or clinical treatments?
- Section 2.3, please add reference to ‘the MCA supplies approximately 80% of cerebral blood flow. Abnormal mCBFV was identified by values < 40 or ≥ 100 cm/sec’.
- The authors used P2/P1 ratio, mCBFV, eICP, PI and eCPP to divide patients into different group with CVH and ICC impairment. How reliable is this evaluation system?
- What software was used for power analysis?
- Methods: invasive ICP detection: please use ‘external ventricular drain’ instead of ‘external ventricular derivation’.
- Methods: How long was each recording last for? 5 minutes? Half an hour? Or one point-measurement?
- Results: Please add a patient flow chart to the results to describe details of how they excluded patients who did not meet the recruitment criteria. In their result section, they mentioned ‘Among those, 1579 were admitted to ICU and 552 critically ill patients died (35%). A total of 50 consecutive patients were eventually included within 72 hours of ICU admission.’ Why the 1579 patients admitted to ICU were excluded? By the way, 2813 (total patient number) – 1579 – 552=682, not 50.
- Results, Table 2: what does * mean? I think the authors already compared the parameters in Table 2 between patients with FO and UO, as well as between survivors and non-survivors. Can they please add the p value into this table to make it ?
- Table 2, Line 22: what does ‘mean arterial pressure < mmHg-no. (%)’ mean?
- What is the definition of hypercapnia in table 2?
- Is there any significant difference in these parameters between the 1st and the 2nd assessment in patients with FO? What about in patients with UO?
- Only CVH/ICC score showed significant difference between FO and UO patients, PI, eICP, eCPP did not show significant difference. Does this mean CVH/ICC is more sensitive and is a better predictor for patient outcome?
- The difference between invasive ICP and non-invasive ICP should be discussed. Different methods for non-invasive ICP assessment have different precision.
Author Response
We would like to thank the editor and reviewers for their valuable comments and insights on our study. We have been working on the manuscript and the final version was clearly improved.
Below our responses are in red, and the manuscript with changes highlighted in yellow is ready for submission. We hope our responses could meet your expectations.
Reviewer #1
1. Covid-19 was used through the whole article, except the last sentence of the introduction section, where they used SARS-CoV-2. Could they make the terminology consistent please?
Authors’ response: The text has been changed, accordingly.
2. Abstract: Please describe the data format in the abstract. For example, 1.20 [1.00-1.28], I assume the data format here is median [95% CI].
Authors’ response: The text has been changed, accordingly.
3. Abstract: The last sentence: ICCI? It should be ICC?
Authors’ response: The text has been changed, accordingly.
4. Delete B4C device in the abstract. It is unnecessary.
Authors’ response: The text has been changed, accordingly.
5. Methods: this study includes data from six intensive ICUs, I assume these are different ICUs? Some are cardiac ICU? Some are Neuro ICU or PCIU? Can they give more details please?
In fact, the entire hospital, with more than 50 surgical rooms and 1000 beds, was entirely dedicated to COVID-19; as such, no more distinction between specialized ICUs was made at that time. This has been added to the text, accordingly.
6. Methods: Can they clarify whether the measurement of TCD or B4C influence clinical decisions or clinical treatments?
As CVH and ICC assessments were not included into therapeutic protocols, these results were not available for clinical decisions or treatments. As such, analysis on outcome was not biased. This has been added to the text, accordingly.
7. Section 2.3, please add reference to ‘the MCA supplies approximately 80% of cerebral blood flow. Abnormal mCBFV was identified by values < 40 or ≥ 100 cm/sec’.
Authors’ response: The text has been changed, accordingly.
8. The authors used P2/P1 ratio, mCBFV, eICP, PI and eCPP to divide patients into different group with CVH and ICC impairment. How reliable is this evaluation system?
The score applied was created for this study, based on previous studies on TCD providing some cut-offs for “abnormal values”. We agree that data on P2/P1 using this new technique are quite limited; this has been added to the text accordingly.
9. What software was used for power analysis?
Authors’ response: We used the assumption made by Kieser and Wassmer on sample size calculation (doi:10.1002/bimj.4710380806).
10. Methods: invasive ICP detection: please use ‘external ventricular drain’ instead of ‘external ventricular derivation’.
Authors’ response: The text has been changed, as requested.
11. Methods: How long was each recording last for? 5 minutes? Half an hour? Or one point-measurement?
There were two sessions of 30 minutes each in average; this has been added to the text.
12. Results: Please add a patient flow chart to the results to describe details of how they excluded patients who did not meet the recruitment criteria. In their result section, they mentioned ‘Among those, 1579 were admitted to ICU and 552 critically ill patients died (35%). A total of 50 consecutive patients were eventually included within 72 hours of ICU admission.’ Why the 1579 patients admitted to ICU were excluded? By the way, 2813 (total patient number) – 1579 – 552=682, not 50.
For our pilot study, and as I was the only trained physician to perform B4C and TCD assessments, we were not able to include a larger number of patients. In this case, the flowchart is less relevant for the study. However, this has been added as a Limitation, accordingly.
13. Results, Table 2: what does * mean? I think the authors already compared the parameters in Table 2 between patients with FO and UO, as well as between survivors and non-survivors. Can they please add the p value into this table to make it?
Authors’ response: We have specified what * stands for. As number of patients in each group was quite limited, almost all analyses revealed non-significant results. In the absence of adjustment or multivariable analyses, and to keep the Table 2 more readable, we prefer not to add all p values for all analyses and report into the text main differences between groups.
14. Table 2, Line 22: what does ‘mean arterial pressure < mmHg-no. (%)’ mean?
Authors’ response: The text has been corrected, accordingly.
15. What is the definition of hypercapnia in table 2?
Authors’ response: This has been added, accordingly.
16. Is there any significant difference in these parameters between the 1st and the 2nd assessment in patients with FO? What about in patients with UO?
Authors’ response: For those with available assessment, no significant differences into the two assessment were observed between groups. This has been briefly added to the text.
17. Only CVH/ICC score showed significant difference between FO and UO patients; PI, eICP, eCPP did not show significant difference. Does this mean CVH/ICC is more sensitive and is a better predictor for patient outcome?
Authors’ response: This is what we have further explained into the manuscript.
18. The difference between invasive ICP and non-invasive ICP should be discussed. Different methods for non-invasive ICP assessment have different precision.
Authors’ response: This has been added, accordingly.
Reviewer 2 Report
Dear Authors, I would congratulate you on this interesting study.
You explored cerebral haemodynamics in COVID-19 patients, a not yet well known field.
I have some minor suggestions.
In the introduction, as it regards neurological manifestations of SARS-CoV2 infection, please see and cite, if you think appropriate, this review: "Deana C, Verriello L, Pauletto G, et al. Insights into neurological dysfunction of critically ill COVID-19 patients. Trends in Anaesthesia & Critical Care. 2021;36:30-38. doi:10.1016/j.tacc.2020.09.005". I think it sums up all you need to take into consideration.
Table 2: please add the number of patients when you talk about female gender (maybe you put in the table only the % and not the absolute value...)
Discussion: second paragraph ("Currently, there are no data on alterations in intracranial hemodynamics... and so on): I personally do not agree with this sentence. As a support of this, please see
1)Battaglini, D. et al. Neurological Complications and Noninvasive Multimodal Neuromonitoring in Critically Ill Mechanically Ventilated COVID-19 Patients. Frontiers in Neurology 11, doi:10.3389/fneur.2020.602114 (2020).
2) Ziai WC, Cho SM, Johansen MC, Ergin B, Bahouth MN. Transcranial Doppler in Acute COVID-19 Infection: Unexpected Associations. Stroke. 2021 Apr 21:STROKEAHA120032150. doi: 10.1161/STROKEAHA.120.032150. Epub ahead of print. PMID: 33878893.
Please take into consideration this two papers.
Finally, I have a curiosity to ask: did you obtain a CSF sample in some of these patients? Did you look for Ig versus SARS-CoV2 in the CSF?
Author Response
We would like to thank the editor and reviewers for their valuable comments and insights on our study. We have been working on the manuscript and the final version was clearly improved.
Below our responses are in red, and the manuscript with changes highlighted in yellow is ready for submission. We hope our responses could meet your expectations.
- In the introduction, as it regards neurological manifestations of SARS-CoV2 infection, please see and cite, if you think appropriate, this review: "Deana C, Verriello L, Pauletto G, et al. Insights into neurological dysfunction of critically ill COVID-19 patients. Trends in Anaesthesia & Critical Care. 2021;36:30-38. doi:10.1016/j.tacc.2020.09.005". I think it sums up all you need to take into consideration.
Authors’ response: Thank you for the suggestion, the reference has been added.
- Table 2: please add the number of patients when you talk about female gender (maybe you put in the table only the % and not the absolute value...)
Authors’ response: This has been added, accordingly.
- Discussion: second paragraph ("Currently, there are no data on alterations in intracranial hemodynamics... and so on): I personally do not agree with this sentence. As a support of this, please see
1)Battaglini, D. et al. Neurological Complications and Noninvasive Multimodal Neuromonitoring in Critically Ill Mechanically Ventilated COVID-19 Patients. Frontiers in Neurology 11, doi:10.3389/fneur.2020.602114 (2020). 2) Ziai WC, Cho SM, Johansen MC, Ergin B, Bahouth MN. Transcranial Doppler in Acute COVID-19 Infection: Unexpected Associations. Stroke. 2021 Apr 21:STROKEAHA120032150. doi: 10.1161/STROKEAHA.120.032150. Epub ahead of print. PMID: 33878893.
Please take into consideration this two papers.
Authors’ response: The manuscript was redacted before these studies were published, thank you for the recommendations.
- Finally, I have a curiosity to ask: did you obtain a CSF sample in some of these patients? Did you look for Ig versus SARS-CoV2 in the CSF?
Authors’ response: Unfortunately, the CSF was not routinely studied, for few cases (only 5) this was obtained, although Ig levels were not measured. This can be taken in consideration for a next study, since the pandemic is still ongoing in our area.
Round 2
Reviewer 2 Report
Thank you, you have addressed all my queries